# Chromosome-Level Genome Assembly and Population Genomic Analyses Reveal Geographic Variation and Population Genetic Structure of *Prunus tenella*

**DOI:** 10.3390/ijms241411735

**Published:** 2023-07-21

**Authors:** Yue Qin, Han Zhao, Hongwei Han, Gaopu Zhu, Zhaoshan Wang, Fangdong Li

**Affiliations:** 1Research Institute of Non-Timber Forestry, Chinese Academy of Forestry, Zhengzhou 450003, China; qinyue870711@163.com (Y.Q.); zhaohan@caf.ac.cn (H.Z.); lifangdong66@163.com (F.L.); 2Economic Forest Research Institute, Xinjiang Academy of Forestry, Urumqi 830000, China; ecoforest@126.com; 3Research Institute of Forestry, Chinese Academy of Forestry, Beijing 100091, China

**Keywords:** *P. tenella*, genome, assembly, almond, population genetic structure

## Abstract

*Prunus tenella* is a rare and precious relict plant in China. It is an important genetic resource for almond improvement and an indispensable material in ecological protection and landscaping. However, the research into molecular breeding and genetic evolution has been severely restricted due to the lack of genome information. In this investigation, we created a chromosome-level genomic pattern of *P. tenella*, 231 Mb in length with a contig N50 of 18.1 Mb by Hi-C techniques and high-accuracy PacBio HiFi sequencing. The present assembly predicted 32,088 protein-coding genes, and an examination of the genome assembly indicated that 94.7% among all assembled transcripts were alignable to the genome assembly; most (97.24%) were functionally annotated. By phylogenomic genome comparison, we found that *P. tenella* is an ancient group that diverged approximately 13.4 million years ago (mya) from 13 additional closely related species and about 6.5 Mya from the cultivated almond. Collinearity analysis revealed that *P. tenella* is highly syntenic and has high sequence conservation with almond and peach. However, this species also exhibits many presence/absence variants. Moreover, a large inversion at the 7588 kb position of chromosome 5 was observed, which may have a significant association with phenotypic traits. Lastly, population genetic structure analysis in eight different populations indicated a high genetic differentiation among the natural distribution of *P. tenella*. This high-quality genome assembly provides critical clues and comprehensive information for the systematic evolution, genetic characteristics, and functional gene research of *P. tenella*. Moreover, it provides a valuable genomic resource for in-depth study in protection, developing, and utilizing *P. tenella* germplasm resources.

## 1. Introduction

*Prunus tenella* is one of the oldest relict plants left from the Tertiary Miocene epoch and is mainly distributed in China and Kazakhstan [1]. In China, *P. tenella* is an endangered and rare species with economic, scientific, and cultural importance and is scattered only in the northern mountainous areas of Xinjiang province [2]. It is considered to be in danger of extinction due to natural and human-caused disturbances, and as a result, is on the list of nationally significant protected wild plants (Figure 1).

As a relatively wild species of cultivated almond, *P. tenella* has adapted to extremely harsh environments, displaying impressive cold and drought tolerance [3]. For example, it can grow normally in the hills and mountains in Tacheng and Altay at extremely low temperatures reaching −35 °C. Additionally, *P. tenella* possesses exceptional agronomic traits and unique genomic characteristics, making it highly valuable for diverse applications in various fields, such as food processing and medicine [2]. Additionally, these characteristics provide precious genetic materials for studying the adaptive evolution of the *P. tenella* genome and allow for the genetic improvement of the cultivated almond, thus improving the resistance of cultivated almond to biotic and abiotic stresses and improving the yield.

With the increased attention to wild resources, many desirable traits have been mined and applied to agricultural varieties [4,5,6,7,8]. Moreover, many breakthrough varieties have also benefited from the discovery of wild-breeding genetic resources [9,10,11,12], especially for wheat, rice, soybean, and other economically important food crops [13,14,15,16]. Together, these lines of evidence have shown that introducing wild gene resources has greatly improved disease resistance, insect resistance, and the growth of cultivated varieties [17,18,19,20,21].

In *P. tenella*, some genes such as self-incompatibility-related genes SBPI, Petcullin1, SFB, SSK1, S-RNase, and the cold resistance-related gene AlsCBF1-A were identified by homologous cloning technology [22,23,24]. Population diversity and structure were studied using molecular markers, including chloroplast DNA (cp DNA), ISSR, and SSR markers. There was a significant genetic structure among the wild almond populations, and the genetic variation mainly came from among the populations. Some regions with high genetic diversity were also found. These studies provide a basis for the distribution, evolution and conservation of wild almond populations [25,26]. However, thorough knowledge of the economically significant genetic features has been severely hampered by a paucity of genomic resources. More research into the evolutionary adaptations and the development process of distinct features is required before the genetic characteristics of *P. tenella* can be properly analyzed.

Obtaining high-quality reference genome sequences is the key to revealing allelic variation, genetic relationship, and evolutionary history [27,28,29,30,31]. The present paper describes a highly accurate chromosome-scale standard genome of *P. tenella* assembled from scratch using the Hi-C technique and lengthy PacBio SMRT reads. Additionally, the genetic variation and geographical differentiation of 130 individual plants from eight wild natural populations were assessed using genome-wide high-resolution molecular markers, which provided an important basis for advancing our understanding of origin, formation, and geographical distribution profile of *P. tenella*.

## 2. Results

### 2.1. De Novo Assembly of the P. tenella Genome

After quality assessment (GC distribution statistics, quality value Q20, Q30 assessment) and filtering, 29.55 Gb clean data were obtained for genome size assessment and assembly. The sequencing depth totaled approximately 137.00×, with a GC content of approximately 38.15%. Additionally, the proportion of Q20 was greater than 97.19%, and the proportion of Q30 was greater than 91.81%. Based on k-mer distribution analysis, the sample genome size was estimated at around 215.65 Mb, consisting of 35.76% repeat sequences and 0.47% heterozygosity. A k-mer distribution of k = 21 was observed (Appendix A).

Using PacBio SMRT single-molecule sequencing, 34.6 GB high-accuracy HiFi reads were obtained for de novo genome assembly. The average length of ccs produced was more than 1765 bp, and the longest was 34,337 bp. After assembly and deduplication, a contig level assembly of 231 M sequences was obtained, a contig-level assembly of 231 M sequences was obtained, comprising 3073 contigs (longest = 35.9 Mb) with a 18.1 Mb contig N50 (Table 1). After sequencing quality control, 36.05 Gb Hi-C Fastq clean data were obtained by 3D de novo assembly with the proportion of Q30 was more than 93.10% and the effect rate of 32.5%. Finally, 231 Mb of sequences were anchored onto eight pseudochromosomes (Figure 2), covering 479 scaffolds (longest = 44.8 Mb, scaffold N50 = 25.6 Mb) (Table 1). A 89.7% mounting ration was estimated (Table 2). The BUSCO results showed that more than 2009 (94.7%) genes could be compared to the lineal homologous database, among which 62.5% were single-copy and 32.7% were duplicates (Table 3).

### 2.2. Functional Annonations, Gene Prediction, and Repetitive Sequences

To further describe the *P. tenella* genome, we categorized all sequences that recur using a combination of de novo and homology-based methods. We estimated that transposable elements made up 28.97% of the genome, with TIRs making up to 7.56% and non-TIRs making up to 3.04%. Only LTR retrotransposons were found, accounting for 18.37%, while non-LTR retrotransposons were not detected (Table 4). The present assembly projected a total of 32,088 protein-coding genes, among which 97.24% had at least one public database functional annotation (Table 5). 

### 2.3. Synteny Analysis, and Genome Evolution and Phylogeny

We utilized 10,184 gene families from related species, including, *P. avium*, *P. armeniaca*, *P. kansuensis*, *P. ferganensis*, *P. dulcis*, *P. davidiana*, *P. humilus*, *P. mandshurica*, *P. mira*, *P. mume*, *P. persica*, *P. salicina*, and *P. sibirica*. Among them, 8674 were common to all species, while 288 were unique to *P. tenella* (Figure 3).

The period of divergence was then approximated by comparing the protein sequences from every single-copy gene families common to those species, and the resulting phylogenetic trees were created. Bootstrap values greater than 90% provided strong evidence for the association between the species (Figure 4). Indeed, all seven species of the peach genus are on the same branch; among them, *P. tenella* is an ancient group that diverged approximately 13.4 million years ago (Mya). Furthermore, *P. tenella* almost diverged about 6.5 Mya from the cultivated almond. A total of 2679 genes involved in expansion and contraction were identified by gene family analysis, among which 1581 genes in 105 gene families were significantly expanded, and 26 genes in 23 gene families were significantly contracted. Together, these data indicate that the *P. tenella* gene family has undergone a significant contraction (2256 genes) relative to other species (486–1613 genes) in the peach genus, which may be related to natural selection.

Next, synteny analysis was performed to further understand genes’ position relationship on homologous chromosomes and the variation of genome structure. Coding gene and genome-wide collinearity analysis showed a highly linear relationship between *P. tenella*, cultivated almond, and peach. Meanwhile, 1,540,264 SNPs, 105,280 deletions, and 155,863 insertions (including absence/presence variations) sequences were identified when compared with almond (Appendix A). Compared with peach, the 1,574,620 SNP, 106,957 deletions, and 161,747 insertions (including absence/presence variations) sequences were identified (Appendix A). These structural variations are the main source of genomic variation and may have a significant association with phenotypic traits. In addition, a large inversion (7588 kb) was identified on chromosome 5, which is valuable for further understanding gene regulation and epigenetic inheritance in *P. tenella* (Figure 5). 

### 2.4. Population Genetic Structure and Genetic Diversity Analysis

Using the Illumina HiSeq X Ten platform, 512.5 Gb of clean PE150 paired-end data (approximately 15 average sequencing depth) were generated from 130 separate specimens gathered from the native area in Xinjiang Province, revealing the genetic variants in *P. tenella* across different geographic populations. Based on the genetic distances derived from the genotypes at all the SNP sites of the eight subpopulations in the three areas, a maximum-likelihood and neighbor-joining (Figure 6A) phylogenetic tree was created using the SNPs/genotypes. Some individuals within the subgroups were grouped in other subgroups, but overall, the groupings from the three locations exhibited strong genetic isolation and constituted distinct groups within the phylogenetic tree. The phylogenetic tree’s conclusion was confirmed by principal component analysis. Tacheng, Yuoli, and Yumin samples clustered together in a distinct subgroup of the PCA (Figure 6B). 

Structure analysis results also showed that the 130 samples were mainly from three ancestral populations, consistent with their distribution areas. The samples from the Tacheng are a pure population, while the TuoLi and Yumin samples are hybridized populations with slight levels of admixture (K = 3; Figure 6C). Genetic diversity analysis showed that Expected_heterozygous_number, Observed_allele_number, Nei_diversity_index, and Shnnon_Wiener_index were 0.17–0.29, 1.31–1.49, 0.18–0.30, and 0.25–0.43, respectively (Appendix A). Compared with other populations, the genetic diversity of the three populations in Yumin was relatively high, while the Tacheng population was relatively low.

### 2.5. Genome-Wide Selection Signatures’ Analysis of Differentiation

To understand the genetic differentiation among populations, we conducted selective sweep analyses and calculated all pairs of Fst values between eight sampling populations in three different regions. The results showed that the genetic differentiation between the Tacheng and Yumin populations was 0.29–0.32. The genetic differentiation between Tacheng and Tuoli was 0.28–0.3, and the genetic differentiation between Tuoli and Yumin was 0.21–0.27. The inter-population genetic differentiation within the region was low, with the differentiation coefficient falling between 0.05 and 0.1 in the four sample populations of Tuoli and between 0.05 and 0.09 in the three sample populations of Yumin. These results indicate that there had been strong genetic differentiation and geographical variation among the different geographical regions of *P. tenella*, which may be due to the obstruction of gene flow caused by geographical isolation.

Based on population differentiation and genomic heterozygosity detection, Tacheng, Tuoli and Yumin regions all experienced strong selection(Figure 7). When Fst ≥ 0.657, the differentiation level between Tacheng and Tuoli populations reached a significant level, among which Tacheng had relatively few selection sites (36) and Tuoli had more selection sites (379), indicating that the Tuoli population had a relatively high selection degree. When Fst ≥ 0.639, the differentiation level between the Tacheng and the Yumin population reached a significant level, and compared with Tacheng population (33), the Yumin population was also more selected in loci 43. When Fst ≥ 0.536, the level of differentiation between the Yumin and Tuoli populations reached a significant level, and the selected sites of the two populations were relatively balanced, ranging from 147 to 153. The analysis of selection sites showed that the selection degree of the Tacheng population was relatively small, and the selection mainly occurred in the Tuoli and Yumin populations.

## 3. Discussion

Through the use of various sequencing methods, we assembled the first complete reference genome for *P. tenella* in this work. The *Prunus genus* has several commercially and ecologically significant species in forestry and agriculture, and these data are essential for learning more about *P. tenella* and the genus as a whole. These findings will also aid in the development of genome-enabled *P. tenella* breeding initiatives. Last but not least, *P. tenella*’s status as a relict species makes it a useful model for studying the genetic basis of population formation, evolution, and adaptation to environmental effects under conditions of geographic isolation.

Given the high quality of our *P. tenella* genome assembly, high-depth PacBio long-read, and whole genome re-sequencing data, we now have a comprehensive understanding of the genome of the *P. tenella*. Single-copy, multi-copy, and species-specific gene families were obtained, the evolutionary status was inferred, and the genome’s evolutionary history was traced, laying the foundation for further exploration and research. Additionally, we found many variation sites, including SNPs, insertions, deletions, and inversion. Many of these variants may be associated with phenotypic traits, which will help understand the phylogenetic evolution of *P. tenella*. Moreover, these various sites can be used as important molecular markers for germplasm identification, genetic analysis, functional gene extraction, and assisted breeding. Compared with the genome of *P. persica* and *P. dulcis*, we identified a large inversion on chromosome 5, which may be related to the unique characteristics of *P. tenella* such as dwarfing and freezing resistance. In order to further determine the authenticity of this inversion, we re-checked the assembly process and the contig connection. Since the chromosome we assembled was composed of two contig and there was a break point at about 7588 kb on one side, we divided 500 kb sequences at both ends of the break point and calculated the degree of linkage disequilibrium between the two connection modes. By calculation, the degree of linkage disequilibrium of the current connection mode is 0.09415, which is significantly greater than that of the other connection mode (0.05116), indicating that our present assembly results are reasonable.

Through the analysis, we also found some different characteristics of the *P. tenella* genome relative to other species of the same genus. Compared with other tree species, *P. tenella* had relatively more endemic gene families (288), while cultivated almond and peach had relatively few, only 31–37, which may reflect the unique evolutionary characteristics of *P. tenella*. More gene families were contracted in *P. tenella*, which might be related to natural selection caused by the extreme natural environment. The evolutionary status inferred by the phylogenetic tree shows that the *P. tenella* were clustered into a large clade with amygdala subgenus, but compared with other species of amygdala subgenus, *P. tenella* differentiated earlier, at 13.4 million years ago, while cultivated almond differentiated only 6.9 million years ago, indicating that *P. tenella* is a relatively old species. But the *P. tenella* is still the closest relative to cultivated almond. Importantly, this observation indicates that *P. tenella* has the potential utilization value of providing genetic resources for cultivated almond, and this lays the foundation for further exploration and research. 

Population genetic structure can be used to analyze the evolutionary dynamics of a population by describing gene transmission, gene frequency change, and genotype distribution [32,33,34,35]. Based on the SNP information derived from the whole-genome re-sequencing data, thousands of single SNP markers can be used for the fine-scale description of genetic structure [36,37,38]. The results showed that, compared with the Tacheng (Nei’s = 0.18; Ho = 0.16) and Tuoli (Nei’s = 0.23–0.26; Ho = 0.16–0.22) populations, the Yumin variety (Nei’s = 0.26–0.3; Ho = 0.17–0.22) has relatively high genetic diversity. This observation is consistent with the study on genetic diversity using chloroplast sequences [4]. Additionally, the results indicated a high genetic differentiation among the natural distribution of *P. tenella*, as the pairwise genetic differentiation (Fst) in a different region is 0.23–0.32, especially within the Tacheng and Yumin group where Fst reached 0.29–0.32, values much higher than wright’s high differentiation coefficient [39,40,41]. However, there is little differentiation between subgroups within the Tuoli and Yumin group (0.05–0.1). These results suggest that geographical isolation is an important factor affecting the genetic evolution of *P. tenella*. This higher differentiation may result from the long-term natural selection without gene flow. 

Selective Sweep analysis showed that the Tacheng population received fewer selection sites, while the Tuoli and Yumin populations received stronger selection sites. This indicates that there is a tendency to decrease genetic polymorphism and increasing purity and degree in Yumin and Tuoli populations. Due to the severe geographical isolation of *P. tenella*, this phenomenon may be related to the small population and inbreeding of the population. Since Tuoli and Yumin are in the same mountain range, and Tacheng is in another mountain range, this conclusion also shows that, under the influence of climatic and geographical conditions, the evolution and selection among the groups are relatively independent, resulting in different evolutionary directions. In summary, we assembled the first chromosome-level genome of *P. tenella* and assessed the genetic variation and geographical differentiation of eight natural populations, which laid a solid foundation for further research on genetic improvement and formation mechanism of important characters in the future.

## 4. Materials and Methods

### 4.1. Utilized Materials

*P. tenella* sample materials used for genome assembly were obtained from the germplasm conservation nursery of Xinjiang Academy of Forestry Sciences, Xinjiang, China. Fresh leaves were utilized for Hi-C library development, PacBio HiFi sequencing, and Illumina sequencing. To aid in genome assembly and annotation, fruit, leaf, root, and stem tissues were taken for RNA-seq study.

The fresh leaves used for whole genome re-sequencing were collected from Yumin County, Tuoli County, and Tacheng City, Xinjiang, China. A total of 8 *P. tenella* populations were collected, including 3 from Yumin County, 4 from Tuoli County, and 1 from Tacheng City (Table 6). Approximately 15–18 samples were collected from each population. In addition, 7 cultivated almond samples were collected for population evolution analysis.

### 4.2. Genome Sequencing and Transcriptome Sequencing

The experiments were carried out in accordance with Illumina’s recommended methodology. The ultrasonic shock was used to physically fragment the qualifying genomic DNA into fragments (350 bp), and then end restoration, adding A, an adapter, and target fragment picking and PCR were used to generate the tiny fragment sequencing library. By using a bridge PCR, the library was transferred to the sequencing chip. An Illumina sequencer performed double-ended 150 bp (PE 150) library sequencing. 

DNA capture and purification, cyclization, end repair, endonuclease digestion, cell cross-linking, and on-machine sequencing were all necessary steps for HI-C sequencing to be completed. The mRNA was utilized to synthesize full-length cDNA with the help of the SMARTerTM PCR cDNA Synthesis Kit, which was then used to generate sequencing libraries. Using the PacBio system, we sequenced the whole transcriptome.

Library sequencing, library quality testing, library creation, and sample quality testing were all carried out as per Illumina’s recommended approach for re-sequencing a variety of population samples. In order to prepare the DNA for sequencing, it was first physically fragmented (using ultrasonic waves), then purified, the ends were mended, the 3’ end was augmented with A, and the sequencing joint was linked. Finally, agarose gel electrophoresis was used to determine the optimal fragment size, and PCR amplification was carried out to form the sequencing library. 

Transcriptome sequencing of the stem, root, leaf, and fruit tissues was performed on the NovaSeq 6000 platform.

### 4.3. Assurance of Sequencing Data Quality

Low-quality sequences and duplicated readings in the sequencing data were removed using stringent filtering algorithms that were optimized for the particular platform utilized to ensure data integrity and accuracy. Filtering criteria included the following actions for Illumina Hi-Seq data: Firstly, polyG tails were removed. Secondly, paired reads of less than 100 bp in length were discarded. Thirdly, read pairs containing more than 10% of bases that are the same as the next base were removed. Fourthly, read pairs with over 50% low-quality bases (quality score less than 10) were discarded. The last step was to clean the data of read pairings with a typical quality rating below 20. The Hi-C sequencing results went through a comparable filtering procedure as Illumina Hi-Seq Data before being processed in 3D. With the default settings of the pbccs pipeline, subreads from the PacBio HiFi long readings were filtered and corrected immediately. Approximately 2000 PacBio HiFi (CCS) reads were randomly selected from the sequencing data and compared with the NT library to evaluate whether the sequencing data contained contamination.

### 4.4. Heterozygosity and Genome Size Estimation

Heterozygosity and genome size were analyzed before HiFi library construction and sequencing. From the Illumina data, Jellyfish v.2.2.10 [42] examined frequency distributions of quality-filtered short fragments (21-mers). Then, based on Jellyfish’s results, genome escope2^2^ was used for genome analysis. This strategy obtained the genomic information of *P. tenella* (Appendix A), such as heterozygosity, genome size, and proportion of repeat sequences.

### 4.5. Genome Assembly

Following correction and filtering, HiFi circular consensus sequencing (CCS) reading could be used in the de novo assembly using hifiasm (v0.14-r312) with default parameters. Purge haplotigs was used to remove redundant haploids [43]. In 2017, Dudchenko et al. [44] used the 3D de novo assembly (3D-DNA) software for scaffolding the haploid contigs. The Hi-C readings could be aligned within the draft genome 3D-DNA and Juicebox v1.9.8 was used for the candidate assembly. Assembly Tools (JBAT) [45] was utilized for reviewing the candidate assembly and corrected artificially. The eudicotyledons_odb10 database was employed in conjunction with the BUSCO v3.0.2 (Benchmarking Universal Single-Copy Orthologs) [46] algorithm for assessing genome integrality and gene annotation. A combination of the BWA-MEM method and HISAT2 (v2.1.0) [47] was utilized for mapping the small reads’ filtration obtained by Illumina and the assembled transcripts to the assembly.

### 4.6. Repetitive Element Annotations

To annotate the TEs or transposable elements [48], the EDTA genome annotation pipeline was utilized. TEs include retrotransposons and DNA transposons. RepeatModeler was used to identify DNA transposons, including long interspersed nuclear elements (LINEs) of the terminal inverted repeats (TIRs) and retrotransposons, and long tandem repeats (LTRs), as well as helitrons found in DNA transposons. To do this, we used Repbase and RepeatMasker (v4.0.7) and Repbase with the optimal settings to generate a de novo repeat library for repeat sequence identification [49,50].

### 4.7. Functional Annonations and Gene Prediction

The StringTie (v1.3.5) and HISAT2 (v2.1.0) pipeline was used for mapping the RNA-seq data within the fruits, leavers, stems, and roots to the genome. Gene prediction together with de novo transcripts assembly were conducted through Trinity [51]. PASA (v2.4.1) pipeline transdecoder4 was also applied to annotate the transcripts-relevant coding regions [52]. Exonerate v2.2.0 carried out homolog predictions. GlimmerHMM (v3.0.4) and the protein sequences of *P. dulcis*, *P. mira*, *P. persica*, *P. armeniaca*, *P. mume,* and *P. salicina* could also be mapped to the genome [53]. For de novo gene speculation, genes from the PASA results were trained by AUGUSTUS (v3.3.3) and SNAP [54,55]. In order to combine the gene models, EVidenceModeler (v1.1.1) was used [56]. The predicted protein sequences were compared to the EuKaryotic Orthologous Groups (KOG), Nr databases, Pfam, SwissProt, Kyoto Encyclopedia of Genes and Genomes (KEGG), and Gene Ontology (GO) to infer possible functions for the protein-coding genes.

### 4.8. Phylogenetic and Gene Family Analysis

Thirteen closely related species were selected for phylogenetic and gene family analysis along with *P.tenella*. Additionally, *P.avium* was selected as the outgroup. The genome database for Rosaceae “www.rosaceae.org (accessed on 11 April 2023)”, and the NCBI database “www.ncbi.nlm.nih.gov (accessed on 5 January 2023)” was used for obtaining the protein sequences of these species. Alignment quality was ensured by excluding sequences with lengths < 100 bp. OrthoFinder (v2.5.2) were deployed for identifying single-copy homologous genes and classifying families, using the settings “-M msa -S diamond -T raxml-ng” [57]. RaxML [58] have been approached for estimating and evaluating the phylogenetic connection tree of 14 species using 100 bootstrap repetitions. Time to diverge was computed using PAML’s MCMC tree [59]. CAFE (v3.1) was used for examining relevant growth patterns and gene families-related declines, as described by Han et al. [60]. By counting the number of ancestral gene families on each branch of the phylogenetic tree, we were able to determine the rate at which gene family sizes shrank or grew. Cafetutorial_clade_and_size_filter.py was used to filter gene families characterized by very high variations in gene copy numbers in an effort to decrease prediction mistakes. Exact data on the contraction and expansion gene families of 14 species were utilized using the script cafetutorial_report_analysis.py, and these data were then analyzed. For selected gene families, we used Fisher’s exact test to analyze GO functional enrichment.

### 4.9. Whole-Genome Synteny Analysis

Almond and peach were selected for whole genome replication (WGD) analysis. Four-fold synonymous (degenerative) third-codon transversion (4DTv) values and synonymous mutation distributions for each synonymous site (Ks) were calculated to analyze the genome replication events. The YN substitution model was used to calculate the 4DTv rates based on four-fold degenerate sites. KaKs_Calculator (v2.0) [61] with default parameters was used to calculate Ks values. The minimap2 software “https://lh3.github.io/minimap2/minimap2.html (accessed on 15 November 2022)” was used for genome-wide comparison, and syri software “https://github.com/schneebergerlab/syri (accessed on 12 November 2022)” was used to identify collinear regions between the two genomes, structural rearrangements (inversion, translocation, and duplication), local variations (SNP, indel, and CNV), and unaligned regions. The nucmer (4.0.0beta2) program in MUMmer4 [62] was used to determine whether similar gene pairs on chromosomal were adjacent in different species.

### 4.10. Single-Nucleotide Polymorphism (SNP) Calling

Trimmomatic v0.36 was used to eliminate adaptors and low-quality sequences during the preprocessing phase. Every sample’s clean reads have been planned using Burrows-Wheeler Aligner to the *P. tenella* standard genome. Next, Picard “http://broadinstitute.github.io/picard/ (accessed on 12 November 2022)” was employed to identify and align the PCR duplicated sample findings. SNP sites in re-sequencing people from diverse geographical regions were identified using GATK v4 (Genome Analysis Toolkit) for SNP recalling. Each genome’s VCF files were generated using variant calling with GATK Hap-lotypeCaller, and then the VCF files for all 137 genomes were combined to create a single VCF file. Only SNPs that had a Hardy–Weinberg equilibrium < 0.001, minor allele frequency > 0.05, and genotype missing rate of 10% for each were kept for further study, narrowing the analysis down to just biallelic variation sites.

### 4.11. Phylogenetic Analysis

A phylogenetic tree was generated using the distance matrix produced by MEGA-CC3.5 software (MEGAX) [63] and 1000 bootstrap repetitions to assess the phylogenetic connection of various individuals in order to study the evolutionary links between different populations. In addition, the SMARTPCA application included in the EIGENSOFT software “https://github.com/chrchang/eigensoft (accessed on 16 November 2022)” was utilized to carry out principal component analysis (PCA) and ascertain the subpopulations’ clustering status [64].

### 4.12. Population Genetic Structure and Genetic Diversity Analysis

In order to learn about the genetic makeup of populations, including their variety, structure, and differentiation, nucleotide diversity was assessed by dividing each population into 10 kb chunks and analyzing a 100 kb window [65]. Using a Bayesian-based strategy, the K-values (the hypothesized number of populations) ranged between 1 and 10 in ADMIXTURE [66]. The optimal K-value was determined using cross-validation statistics across five separate studies. Bar graphs of the Q matrix for each K-value were made with the aid of the R package Pophelper “http://royfrancis.github.io/pophelper (accessed on 25 November 2022)”. The fixation index (FST) and nucleotide diversity ratios (π) were computed using VCF methods to identify genomic areas possibly experiencing natural selection sweeps throughout the adaptation process.

### 4.13. Selective Sweep Analysis

Genome-wide detection on selective sweep region was processed by calculating the population genetic index of all SNPs within a sliding window of 100 kb and a certain step (10 kb). The indicators include population differentiation fixation index (Fst) and nucleotide polymorphism (π). The index was calculated by the PopGenome package based on consensus SNPs with a pre-defined bin and step.

## Figures and Tables

**Figure 1 ijms-24-11735-f001:**
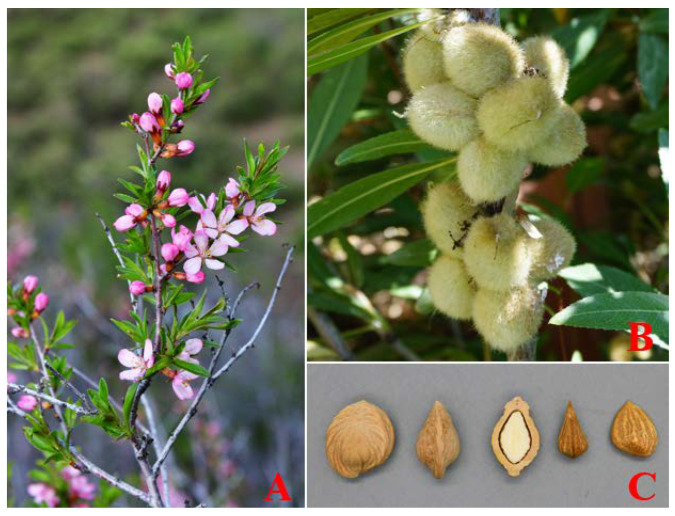
Morphological characteristics of the *P. tenella*: *(***A**) flower; (**B**) fruit; and (**C**) seed.

**Figure 2 ijms-24-11735-f002:**
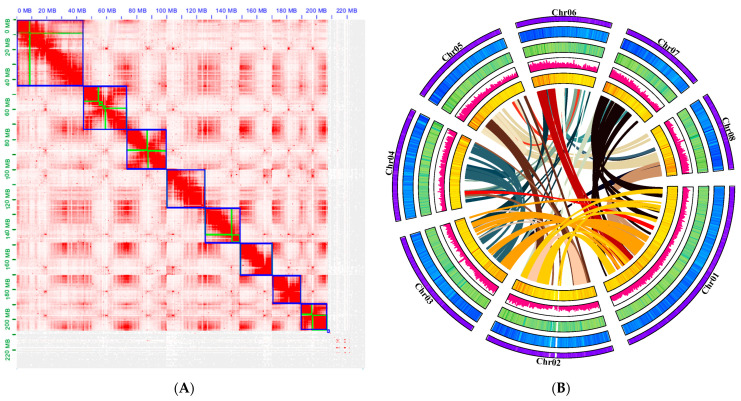
Hi–C assisted assembly of *P. tenella* genome pseudomolecules: (**A**) High-resolution (100 kb) Hi-C interaction heat map among eight chromosomes. The regions in the blue box represent chromosomes and the green within the box represents contig; (**B**) Chromosome characteristics of the *P. tenella* genome. The following items follow the outermost ring: The pseudochromosomal visualization of gene density, GC content, repeat content, SNP density, and gene collinearity from a genome assembly.

**Figure 3 ijms-24-11735-f003:**
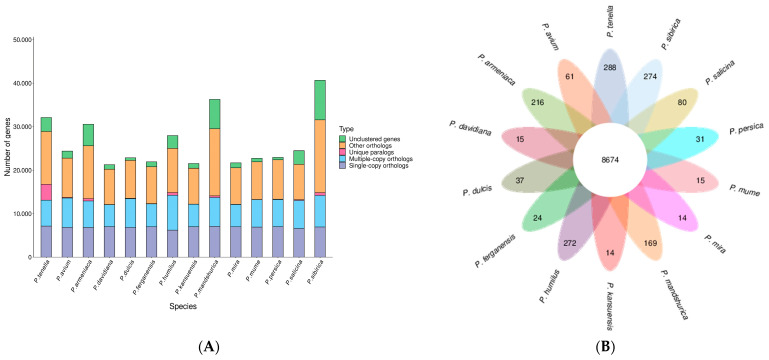
Number of homologous genes in the species and gene family clustering: (**A**) Homologous gene statistics; (**B**) Gene family clustering petal map.

**Figure 4 ijms-24-11735-f004:**
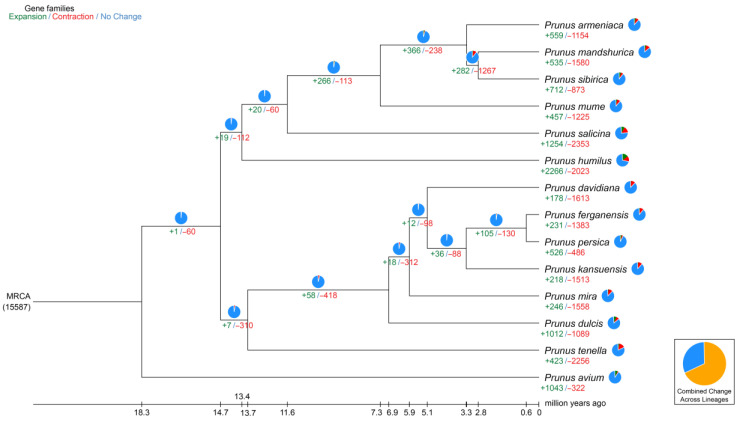
Divergence time and phylogenetic correlation among species. The percentages of conserved (blue), contracted (red), and expanded (green) gene families among all gene families in the 14 species is shown as a pie chart.

**Figure 5 ijms-24-11735-f005:**
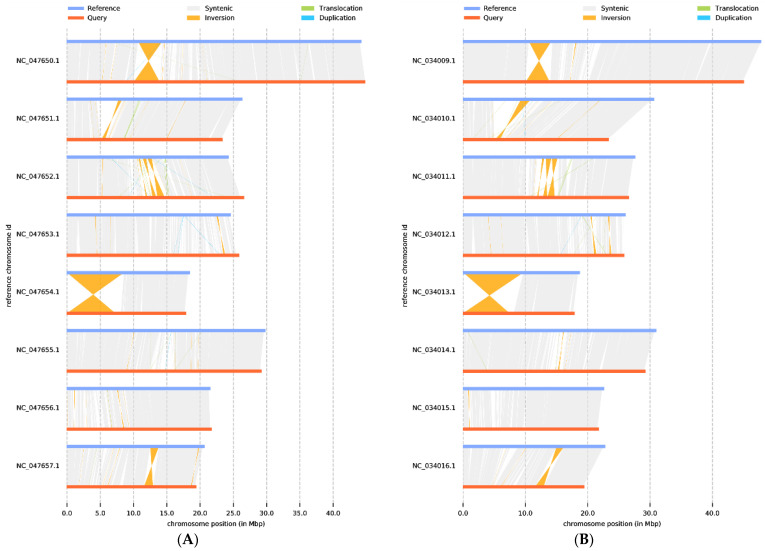
Genome structure map: (**A**) *P. dulcis* vs. *P. tenella*, (**B**) *P. persica* vs. *P. tenella*. The grey area is the collinear relationship area, yellow is the inversion area, green is the translocation area, and blue is the area where reduplication occurs.

**Figure 6 ijms-24-11735-f006:**
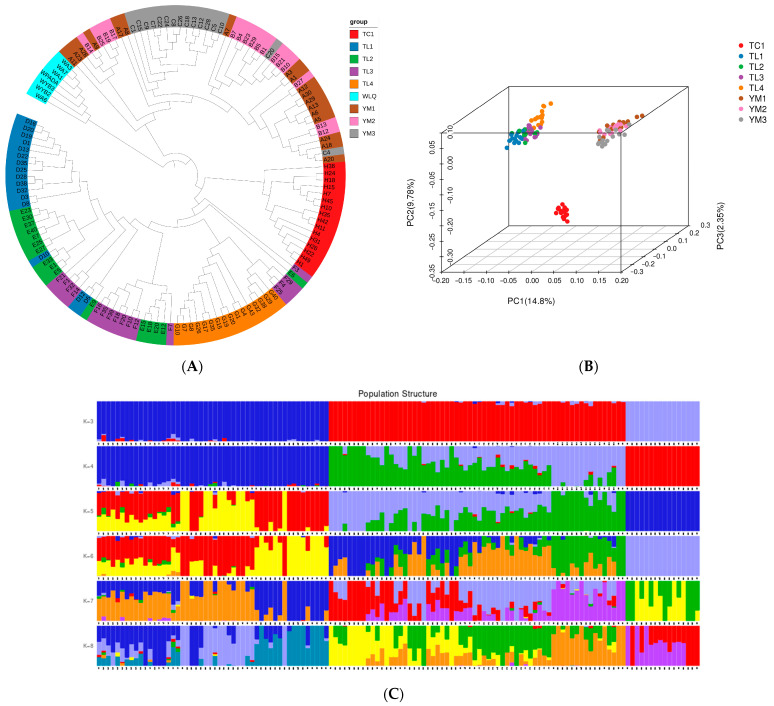
Phylogeny and population structure of different populations. (**A**) Phylogenetic tree; (**B**) PCA three-dimensional cluster diagram of samples; and (**C**) Admixture sample clustering results corresponding to K values (3–8). The length of the different color segments indicates the proportion of a particular ancestor in the individual’s genome.

**Figure 7 ijms-24-11735-f007:**
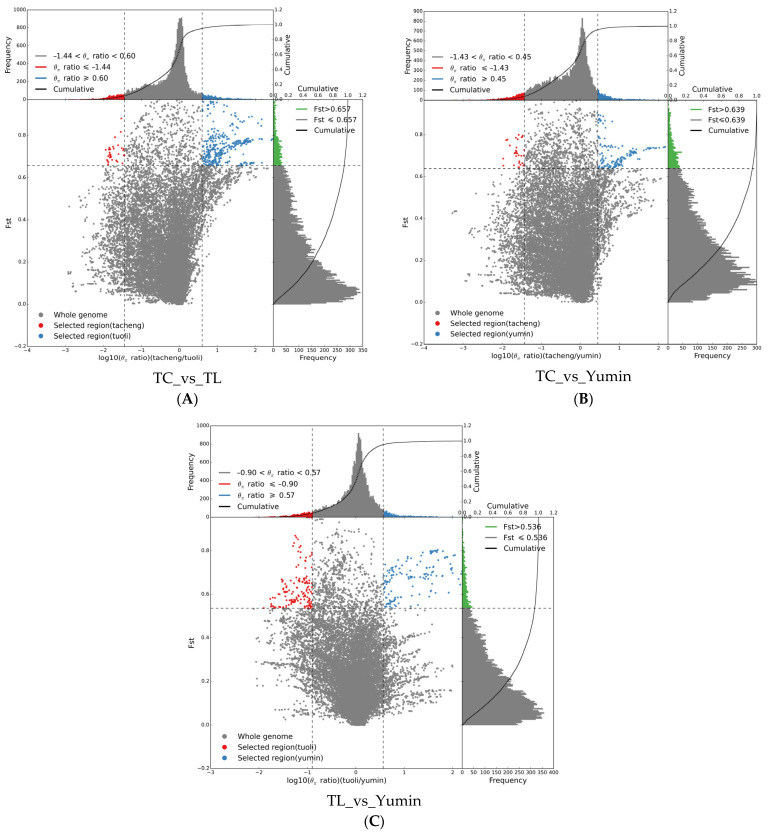
Schematic of selection signatures between populations: (**A**) Selective sweep analysis of Tacheng vs. Tuoli; (**B**) Selective sweep analysis of Tacheng vs. Yumin; and (**C**) Selective sweep analysis of Tuoli vs. Yumin. The ratio of π in the abscissa and Fst in the ordinate corresponding to the frequency distribution diagram above and the frequency distribution diagram on the right, respectively. The dot plot in the middle represents the corresponding Fst and π ratios in different windows. The blue and red areas at the top are the top 5% of the region selected by π, the green area at the right is the top 5% of the region selected by Fst, and the blue and red areas in the middle are the intersection of Fst and π, which are the candidate sites.

**Table 1 ijms-24-11735-t001:** Statistical findings of the assembled *P. tenella* genome.

Term	Contig Number	Contig Size (bp)	Scaffold Number	Scaffold Size (bp)
N90	5	270,515	9	1,233,122
N80	12	8,938,573	7	19,226,261
N70	11	10,302,541	6	21,538,263
N60	15	14,010,243	5	23,141,640
N50	1	18,100,976	4	25,637,364
Max length (bp)		35,886,393		44,825,466
Total size (bp)		231,191,648		231,208,648
Total number		513		479
Average length		450,665.98		482,690.29
Number ≥ 10 kb		513		479

**Table 2 ijms-24-11735-t002:** Statistical findings of chromosomal level assembly of *P. tenella*.

Chr ID	Length (bp)
Chr1	44,825,466
Chr2	29,024,987
Chr3	26,387,986
Chr4	25,637,364
Chr5	23,141,640
Chr6	21,538,263
Chr7	19,226,261
Chr8	17,664,456
Total chromosome level scaffold length	207,446,423
Total scaffold length	231,208,648
Chromosome length/total length	89.7%

**Table 3 ijms-24-11735-t003:** Findings of *P. tenella* genome integrality assessment by BUSCO.

Library	Eudicotyledons_odb10
Fragmented BUSCOs (F)	48
Missing BUSCOs (M)	64
Complete and duplicated BUSCOs (D)	42
Complete and single-copy BUSCOs (S)	1967
Complete BUSCOs (C)	2009
Total BUSCO groups searched	2121
Summary	94.7%

**Table 4 ijms-24-11735-t004:** Statistical findings of repeated sequences TE annotations in *P. tenella* genome.

Class	Length (bp)	Type	Sub-Class	(%)
Retrotransposons	8,454,829	Ty1/Copia	LTR	3.66%
	18,247,107	Ty3/Gypsy		7.89%
	15,756,928	unknown		6.82%
	-	LINE	Non-LTR	-
	-	Unknown		-
DNA transposons	1,845,596	CACTA	TIR	0.80%
	7,818,414	Mutator		3.38%
	4,181,524	PIF/Harbinger		1.81%
	278,071	Tc1/Mariner		0.12%
	3,345,229	hAT		1.45%
	7,038,992	Helitron	Non-TIR	3.04%
Total	66,966,690			28.97%

**Table 5 ijms-24-11735-t005:** Protein-coding genes-related functional annotations in *P. tenella* genome.

Database	Gene Numbers	(%)
GO	10,761	33.54
KEGG	11,435	35.64
KOG	19,251	59.99
Swissprot	19,449	60.61
Pfam annotation	22,815	71.1
Nr annotation	31,202	97.24

**Table 6 ijms-24-11735-t006:** Latitude and longitude information of the sample population.

No.	Population	Location	Latitude	Longitude
1	Yumin 1	Yumin county	45.89251967° E	82.52098163° N
2	Yumin 2	Yumin county	45.90577291° E	82.51545937° N
3	Yumin 3	Yumin county	45.91605329° E	82.50676816° N
4	Tuoli 1	Tuoli county	46.15794124° E	83.56189391° N
5	Tuoli 2	Tuoli county	46.15213502° E	83.53771499° N
6	Tuoli 3	Tuoli county	46.13930633° E	83.55344239° N
7	Tuoli 4	Tuoli county	46.14667251° E	83.57180015° N
8	Tacheng 1	Tahcheng City	46.95030549° E	83.20683721° N

## Data Availability

The whole genome sequence data reported in this paper were deposited in the Genome Warehouse in National Genomics Data Center, Beijing Institute of Genomics, Chinese Academy of Sciences/China National Center for Bioinformation, under accession number GWHCBGA00000000 that is publicly accessible at https://ngdc.cncb.ac.cn/gwh. The datasets generated and analyzed during the current study are available from the corresponding author on reasonable request.

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
