# Peer review of "Chromosome-Level Genome Assembly and Population Genomic Analyses Reveal Geographic Variation and Population Genetic Structure of Prunus tenella"

_ijms, 2023, doi:10.3390/ijms241411735_

Round 1

Reviewer 1 Report

The manuscript entitled “Chromosome-level genome assembly and population genomic analyses reveal Geographic variation and population genetic structure of Prunus tenella” describes chromosome-scale standard genome of P. tenella assembled from scratch using Hi-C technique and lengthy PacBio SMRT reads. Additionally, the genetic variation and geographical differentiation of 130 individual plants from 8 wild natural populations were assessed using genome-wide high-resolution molecular markers. From a scientific point of view, an interesting study with relevant data and results with great analyses has been done however for more clarification I just leave the following comments:

1- Why geographical start with the capital “G”? Please write the scientific name in Italic.

2- In introduction Prunus tenella has introduced as wild almond while there are some othe species within Prunus and even other genera from Rosaceae family as wild almond so it is better to write just scientific name.

3- In follow to the previous question, is P. tenella crossable with Prunus amygdalus or it is used as a rootstock?

4- I think the introduction needs to be more expanded by including more information about the target species, and even the used method for sequencing, and last but not least the importance of population structure analysis in breeding programs.

5- However the manuscript is well-written it needs a moderate English polishing for potential grammatical and structural errors throughout all parts of manuscript. As in some section, the sentences lack logical structure or lack relevant verbs.

6- Please provide all exploited information in material and methods even default parameters of used software. This could include Gene ID of the used sequence, gap penalty and gap extension used, the exact version of the software (in case of online tools mention access date), and other cases.

7- Please provide figures with higher quality as the presented figures are blurred and it is hard to see the result and interpretation. Moreover, there is “alt + a plus a Chinese word” inside Fig 6C which should be eliminated.

8- Why in comparison of genomic results with those Prunus species and also divergence and phylogenetic analysis P. amygdalus is absent?

Altogether manuscript is interesting and I rate it as strong and providing minor revision I strongly recommend it for publication.

However, the manuscript is well-written it needs moderate English polishing for potential grammatical and structural errors throughout all parts of the manuscript. As in some sections, the sentences lack logical structure or lack relevant verbs.

Reviewer 2 Report

The authors constructed pseudomolecules of Prunus tenella, that is endangered and one of the important materials for breeding programs of Prunus species. Using 8 populations from Yumin, Tuoli and Tacheng and resensencing data of those populations, authors clarified that those populations were genetically differentiated based on the the locations.

The volume of data including long and short reads are formidable. On the other hand, explanations of Tables and Figs in the main text are not enough.

Title, Prunus tenella should be italic.

Line 40, The sentence "a relatively wild species of cultivated almond, wild almond" is confusing.

Line 58, S-RNase

Line 89, Table 1 

Fig3A The resolution of the fig is too low.

Fig3A,B,SUP Table 1 and 2, probably "P.tenella" instead of "P.nana"

Line 117, The sentence "Table 4 shows that only long terminal repeat (LTR) retrotransposons were found." is not clear. Provide precise descriptions of Table4, including non LTR and DNA transposons.

Table2 Total chromosome level scaffold length instead of "Total chromosome level contig length"

Line 276, Please check the order of Tables and Figs. Figure 1 should come before other Figs.

Line 281, Information of locations such as latitude and longitude is necessary for population analysis。

Line 328, circular consensus sequencing (CCS)

Line 402, Hap-lotypeCaller

Line 424 What is the VCF methods, any software and citation?

Figure 7. There is no description for the Fig 7 in the result. I suppose the regions exhibited strong selective sweeps are important for understanding history of populations. Also, how authors estimated the selective sweep should be added to material and methods. 

Figure 2,5 and 6A, Which software or R package did authors use for Fig construction ?

Discussion is too short, although authors have sufficient volumes of data to discuss. Authors can add a discussion of the large inversion, comparison of divergent time of other studies, the result of selective sweep and so on.

Reviewer 3 Report

I appreciate this opportunity given to me for reviewing this manuscript. The manuscript “Chromosome-level genome assembly and population genomic analyses reveal Geographic variation and population genetic structure of Prunus tenella” submitted by Qin et al. assembled the first chromosome-level genome of P. tenella, assessed the genetic variation and geographical differentiation which laid a foundation for further research on genetic improvement. The research approach is not sound. No biological or technical replicates were used in the study. The manuscript lacks a proper discussion of the results.

Major issue:

1.     Materials and methods: utilized materials section: What was the reason for selecting those specific samples?

2.     Where are the biological and technical replicates for genome and transcriptome sequencing?

3.     How was the DNA quality assessed?

4.     The findings are not discussed properly.

5.     Figure 3 and Figure 7 are not clear.

6.     What is (Alt+A) inside Figure 6?

Minor issue:

1.     The citation order of 2 and 3 should interchange.

2.     Line # 288: How genomic fragments of size 350bp were ensured?

3.     It is not clear why both Illumina and PacBio systems were used. PacBio normally provides a platform for long-read sequencing but there is mention of why this approach could be useful in the current research.

Minor editing of the English language is required.

Round 2

Reviewer 3 Report

The authors have not revised the manuscript properly. The rebuttal was not convincing. No line numbers were indicated in the rebuttal for the revisions made. The authors should discuss all the results in detail, it's not only about the large inversion, divergent time, and selective sweep. Authors should introduce the clarifications in the revised manuscript wherever appropriate.

Minor editing required.
